# Beyond Guidelines and Reports on Bacterial Co-/Superinfections in the Context of COVID-19: Why Uniformity Matters

**DOI:** 10.3390/antibiotics11101446

**Published:** 2022-10-20

**Authors:** Johan Van Laethem, Denis Piérard, Sabine D. Allard

**Affiliations:** 1Department of Internal Medicine, Universitair Ziekenhuis Brussel (UZ Brussel), Vrije Universiteit Brussel (VUB), 1090 Brussels, Belgium; 2Microbiology Department, Universitair Ziekenhuis Brussel (UZ Brussel), 1090 Brussels, Belgium

**Keywords:** bacterial co-infection, bacterial superinfection, antibiotic stewardship, COVID-19

## Abstract

Background: In the period following the declaration of the COVID-19 pandemic, more evidence became available on the epidemiology of bacterial co-/superinfections (bCSs) in hospitalized COVID-19 patients. Various European therapeutic guidelines were published, including guidance on rational antibiotic use. Methods: In this letter to the editor, we provide an overview of the largest meta-analyses or prospective studies reporting on bCS rates in COVID-19 patients and discuss why the reader should interpret the results of those reports with care. Moreover, we compare different national and international COVID-19 therapeutic guidelines from countries of the European Union. Specific attention is paid to guidance dedicated to rational antibiotic use. Results: We found a significant heterogeneity in studies reporting on the epidemiology of bCSs in COVID-19 patients. Moreover, European national and international guidelines differ strongly from each other, especially with regard to the content and extent of antibiotic guidance in hospitalized COVID-19 patients. Conclusion: A standardized way of reporting on bCSs and uniform European guidelines on rational antibiotic use in COVID-19 patients are crucial for antimicrobial stewardship teams to halt unnecessary antibiotic use in the COVID-19 setting.

## 1. Introduction

The emergence of multidrug-resistant (MDR) bacterial infections has resulted in scarifying future projections. A report from the World Health Organization (WHO) labeled the problem as “so serious, that it threatens the achievements of modern medicine” [1]. For the last few decades, local, national and international efforts have been endorsed by scientific and public health organizations, governments and caregivers to halt the emergence of antimicrobial resistance (AMR). Amongst other interventions, antimicrobial stewardship (AST) teams became the standard of care, guidelines and scientific publications encouraging rational antibiotic use were published, national antibiotic action plans were launched and awareness was cultivated in the general population.

In this perspective article, we first provide a summary of the impact of the COVID-19 pandemic on antimicrobial stewardship efforts, AMR and the growing knowledge on bacterial co-/superinfection epidemiology and antibiotic (over)use in this context. Second, we offer a critical analysis of the major papers reporting on bacterial co-/superinfection (bCS) rates in COVID-19 patients. Last, we discuss the variation in European guidelines for the diagnosis/treatment of these bCSs.

## 2. Antimicrobial Stewardship Applied to COVID-19 Patients: The Pursuit of Knowledge

At the time the COVID-19 pandemic emerged, AST teams and other actors within the healthcare system constrainedly invested great amounts of time and resources in the contention of the pandemic, the procuration of protective equipment and the reorganization of the healthcare system. The latter inevitably resulted in less stringent antimicrobial stewardship, leaving the battle against AMR in the background. Currently, there is insufficient evidence that the COVID-19 pandemic fueled the AMR threat, as present reports are context-specific and differ geographically. However, there are some ominous signs of increased AMR since the emergence of the COVID-19 pandemic. For the European Union, the European Antimicrobial Resistance Surveillance (EARS-Net) network reported a significant rise in carbapenem-resistant *Enterobacterales*, *Pseudomonas aeruginosa* and *Acinetobacter* species as well as vancomycin-resistant enterococci for the year 2020 [2]. Moreover, there was a significant rise in carbapenem use in that same year. The rise in MDR pathogens was most marked in the intensive care setting [3].

In the period following the declaration of the COVID-19 pandemic by the WHO on 11 March 2020, many admitted COVID-19 patients empirically received antibiotics [3,4]. This period of “antibiotic anarchy” can (partly) be explained by the lack of knowledge and lack of guidelines concerning the epidemiology and treatment of presumed bCSs in the context of COVID-19. From May 2020 onwards, the first reports and meta-analyses regarding the incidence and prevalence of bCSs showed very low rates of bacterial co-infections (2.2–8%) and low rates of bacterial superinfections (2.2–20%) in admitted COVID-19 patients. This was in contrast to disproportionally high antibiotic prescribing rates (up to 85%) [4,5,6,7,8]. Soon after the first published reports on bCS incidence, guidelines on the management of presumed bCSs in COVID-19 patients were published by the European Society of Clinical Microbiology and Infectious Disease (ESCMID) (in April 2020) and the WHO (May 2020) [9,10]. Later on, evidence on the good negative predictive value of procalcitonin in excluding bCSs in the context of COVID-19 became available [11,12]. Consequently, over time, a learning effect and a decrease in antibiotic use were noted in some studies [13]. However, antibiotic overprescribing and low quality of antibiotic prescriptions are still prevalent in admitted COVID-19 patients [14].

## 3. Evidence on bCS Rates in COVID-19 Patients: A Critical Point of View

Robust scientific evidence on COVID-19-related bCS epidemiology and clear therapeutic guidance regarding antibiotic use in COVID-19 patients are crucial for antimicrobial stewardship (AST) teams to prevent antibiotic overprescribing. Despite all progress made to gain expertise on bCS prevalence and antibiotic prescribing patterns in COVID-19 patients, significant knowledge gaps and flaws prevail. Moreover, practical guidance on judicious antibiotic use in COVID-19 patients should be improved, as these guidelines are very heterogeneous and lack specificity. Finally yet importantly, meta-analyses and prospective studies reporting on bCS rates in COVID-19 patients (Table 1) should be interpreted with care for several reasons.

First, there is a significant heterogeneity in the used definitions of co-infection and superinfection (also referred to as “secondary infection”). Certain reports use the term co-infection as “every infection contracted before or during the first 48 h of admission”, while others use 24 h of admission as a time limit to differentiate co-infection from superinfection. Some even refer to “every infection diagnosed on presentation”, while in the meta-analysis of Langford et al. (2022) [5], no definition of co-infection is mentioned. Depending on the used definition, bacterial co-infection and superinfection rates can thus be under- or overestimated. Moreover, international COVID-19 therapeutic guidelines such as the World Health Organization and the European Centre for Disease Control and Prevention guidelines do not define bacterial co-infection and superinfection [10]. Second, although most studies exclusively included microbiological diagnoses, it is not always clear if clinical diagnoses, based on other criteria than microbiological documentation, were included. While the first meta-analysis of Langford et al. (2020) [4] did not mention if included diagnoses were exclusively based on microbiological criteria, their second meta-analysis stated that “presumed” or “suspected” diagnoses of infection were excluded. Depending on the used definition of “infection”, the final bCS rate will be different. Third, there is also an important heterogeneity regarding the included microbiological diagnoses. For example, the meta-analysis of Lansburry et al. reports high rates of *Mycoplasma pneumoniae* infections (representing 42% of all reported bacterial infections) [6]. Although no information is provided on the used diagnostic methods, this could be an overestimation due to the inclusion of patients with aspecific serological results. The ISARIC study did not include any *Mycoplasma pneumoniae* infection, as routine testing for atypical pathogens was discontinued in most United Kingdom laboratories during the study period. Fourth, the proportion of included patients depending on the setting (ward versus ICU) and age (adult versus pediatric patients) is not always clearly mentioned. Fifth, some reports include viral and fungal co-/superinfections together with bCSs. Last, the reported infection sites can differ from one study to another. Although most studies focused on both respiratory and bloodstream infections, some also included urinary tract infections whereas others exclusively included respiratory tract co-/superinfections. One should thus pay attention to the reported endpoint when comparing different studies.

## 4. European Therapeutic COVID-19 Guidelines: An Emphasis on Antibiotic Guidance

As the COVID-19 pandemic progressed, knowledge about bCS epidemiology and antibiotic prescribing in COVID-19 patients increased rapidly. This led to the publication of various antibiotic guidance guidelines. However, studies reporting on bCS rates and guidelines regarding judicious antibiotic use show a significant heterogeneity.

Therefore, we analyzed and compared all published national and international guidelines on COVID-19 therapeutic guidance in the European Union (EU) (see Appendix A for methods, complete data and references). Most countries have published their own national therapeutic guideline (see Figure 1), while others refer to international scientific guidelines, such as those of the World Health Organization [10]. Certain countries, such as Austria, refer to the guidelines of neighboring countries. The majority of the EU countries have also included specific guidance on rational antibiotic use in COVID-19 patients. However, there is a large variability in the extent and content of provided guidance regarding antibiotic use in this setting (Appendix A). For example, the Dutch Working Party on Antibiotic Policy, as well as the health authorities of Bulgaria, dedicated specific attention to rational antibiotic use in the COVID-19 setting, while in the national guidelines from other countries, such as Belgium, France, Italy, Poland and Spain, only a few sentences on rational AB are found [15,16,17,18,19,20,21]. While bacterial co-infections are rare in admitted COVID-19 patients, secondary infections are more prevalent in patients with severe COVID-19. This is probably why most guidelines recommend initiating empiric antibiotics exclusively in patients with severe infection, provided that the need for antibiotics would be regularly evaluated. This is in contrast with the Polish national guidelines, which strongly advise against antibiotics in cases of acute respiratory distress syndrome (ARDS) unless there are evident signs of secondary bacterial infection [20]. Some guidelines limit themselves to recommending antibiotics in cases of suspected bCSs, without elaborating on how to diagnose bCSs [19,20]. The ESCMID guidelines state that only patients with clinical or radiological suspicion of bacterial co-/superinfection should receive empirical antibiotics [9]. However, this is quite vague, as radiological consolidations and clinical signs, such as fever and elevated inflammatory markers, are often present in the context of COVID-19. Therefore, the ECDC guidelines advocate for more clarity in defining secondary bacterial infections in COVID-19 patients [21]. Although the Croatian guidelines advocate that bacterial infection is likely in case of leukocytosis and/or a neutrophil left shift with increased procalcitonin concentration and very high CRP and IL-6 levels, elevated procalcitonin and IL-6 levels have low positive predictive value for bacterial infection in the COVID-19 setting and are also observed in the context of COVID-19 sepsis. Those same guidelines, together with the Danish guidelines, suggest following the “sepsis campaign” guidelines in cases of COVID-19 sepsis [22,23]. However, as COVID-19 sepsis is due to a hyperinflammatory state with a potential cytokine storm, this does not necessarily reflect bacterial sepsis. Yet most sepsis campaign guidelines focus on bacterial sepsis, and this includes the empiric use of antibiotics. Nevertheless, one could agree to empirically start antibiotics in severe and degrading presentations of COVID-19. While the German COVID-19 guidelines recommend antibiotic prescribing at admission in the intensive care unit, those same guidelines paradoxically state that there is no place for prophylactic antibiotics [24,25]. Despite the good negative predictive value of low procalcitonin levels for bCSs [11,12], the use of this predictor is only incorporated in the Latvian guidelines [26].

## 5. Conclusions

In conclusion, studies reporting on bCS rates and guidelines regarding judicious antibiotic use show significant heterogeneity. Antibiotic prescribing guidelines depend too much on clinical judgment and should instead take variables into account that have proven to be good predictors or excluders of bCSs, such as procalcitonin. The roles of other potential markers and predictors of bCSs, such as certain comorbidities, the presence of immune suppression and the presence of dense radiological consolidations, are still unclear and should be further investigated.

A standardized way of reporting on bCSs in the context of COVID-19 is the only way to obtain more robust and precise evidence on their incidence and associated risk factors. We therefore strongly advocate for the implementation of international diagnostic guidelines, using predictors and excluders of bCSs and standardized definitions of bCSs. These definitions and guidelines should be dynamic and more detailed. Guidelines should be based on different clinical situations and could indicate the level of diagnostic certainty, and they should evolve according to the best available evidence.

## Figures and Tables

**Figure 1 antibiotics-11-01446-f001:**
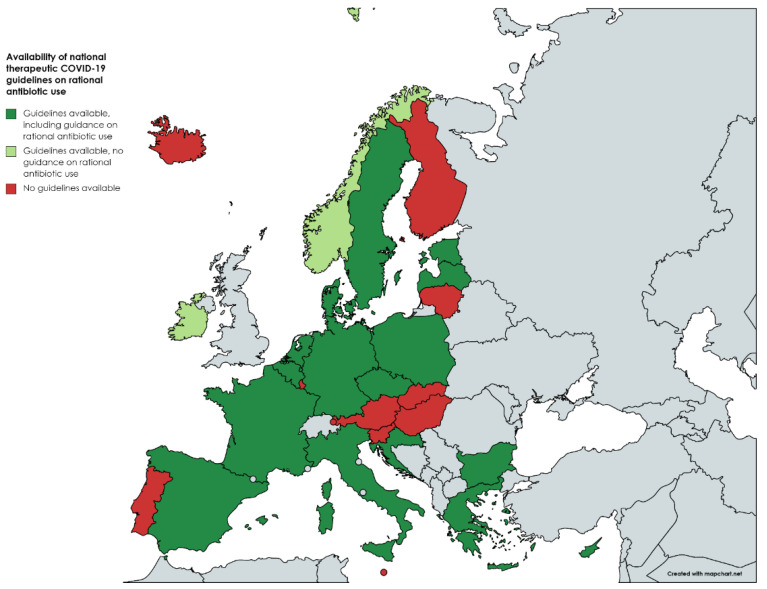
Availability of national therapeutic COVID-19 guidelines on rational antibiotic use. Created with Mapchart: https://www.mapchart.net/europe.html; accessed on 10 October 2022.

**Table 1 antibiotics-11-01446-t001:** Overview of largest (n > 3000) meta-analyses or prospective studies reporting on bacterial co-/superinfection rates in COVID-19 patients (see Appendix A for the search strategy).

Reference and Type of Study	Co-/Superinfection * Definitions	Used Diagnostic Criteria	Reported Pathogens	Reported Infections	Setting (Ward/ICU)	Age Group	Co-/Superinfection Rate	Antibiotic Prescription Rate
Langford et al. (2020) [4] Systematic meta-analysis	co-infection: “on presentation” Superinfection: “emerging during the course of illness or during hospitalization”	Not mentioned if clinical and/or microbiological diagnosis	Bacterial	Respiratory tract infections and bloodstream infections	Ward and ICU	Pediatric and adult patients (25%/75%)	co-infection 3.5% superinfection 14.3%	72%
Langford et al. (2022) [5] Systematic meta-analysis	co-infection: not defined	Microbiological diagnosis Exclusion of “presumed” or “suspected” bacterial infection	Bacterial	Respiratory tract infections and bloodstream infections	Ward and ICU	Pediatric and adult patients	co-infection 5.1% secondary infection 13.1%	75%
Lansburry et al. [6] Systematic meta-analysis	co-infection: not defined. Unclear if this term was used to group “co-infections” and “superinfections”	Microbiological diagnosis (culture and PCR)	Bacterial, viral, fungal	Respiratory tract infections and bloodstream infections	Ward and ICU	Pediatric and adult patients	co-infection 7% (bacterial)	NR
Musuuza et al. [7] Systematic meta-analysis	co-infection: “at the time of a SARS-CoV-2 infection” superinfection: “during care for SARS-CoV-2 infection”	Microbiological diagnosis	Bacterial, viral, fungal	Respiratory tract infections	Ward and ICU	Pediatric and adult patients	co-infection 8% superinfection 20% (bacterial)	NR
Russell et al. [8] Original paper	Co-infection: clinically significant positive results from samples collected within 2 days of admission Superinfection: infection occurring > 2 days after hospital admission	Microbiological diagnosis	Bacterial, fungal	Respiratory tract infections and bloodstream infections	Ward and ICU	Not reported	co-infection 0.7% “secondary” infection 1.5%	85%

* Superinfection and secondary infection are used as synonyms. NR: not reported.

## Data Availability

See Appendix A for additional data.

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
