# Peer review of "Beyond Guidelines and Reports on Bacterial Co-/Superinfections in the Context of COVID-19: Why Uniformity Matters"

_antibiotics, 2022, doi:10.3390/antibiotics11101446_

Round 1

Reviewer 1 Report

I read this letter authored by Laethem et al. and entitled (Beyond guidelines and reports on bacterial co-/superinfections in the context of COVID-19: why uniformity matters). The manuscript is well-presented and describes a current problem usually seen during outbreaks and pandemics. The authors smoothly moved from the introduction of the topic to the epidemiology of bacterial coinfection and superinfection which greatly varied and this variation is well explained by the lack of solid definitions and different diagnostic methods used in the clinical settings. Then, the authors discussed the results and concluded this manuscript with the importance of uniformity of recommendations regarding COVID-19 co-infection and superinfection. In the supplementary materials, the authors described the methodology of this study and summarized national guidelines on COVID-19 therapeutic guidance in the European Union. 

I have no major comments as the article is well articulated and the current recommendations should be considered by official bodies in the concerned regions. However, the areas of improvement for this manuscript could be summarized in the followings:

# More elaborations are preferred on the methodology of table 1, using the PRISMA method for example, when studies are limited to only 4 reviews or original studies, and this could be attached as supplementary materials.

# In table 1, another column deemed important to mention the type of study, i.e., either original or literature review.

# In table 1, in Russel et al study, it was not mentioned the rate of co-infection or superinfection. This cell should be unified with others.  

# I would love to read what other nations in the east have done regarding this problem, especially in China, as the pandemic originated there.

# The part about variation in the definitions of bacterial coinfection and superinfection seems very interesting to read. It could be completed by adding a table of what international and trusted health officials (e.g., WHO. CDC. ECDC.. etc) have defined those terms. 

# Figure 2: the title is a repetition of the footnotes and either one might be enough, and the figure could be moved to the supplementary materials as it is complementary to Table S1. 

# The manuscript needs to be revised as it has some minor grammar and spelling mistakes, (for example: see line 143 "admisson >> admission??) 

If these comments are considered by the authors. this manuscript deserves to be benefited from the second round of revision. 

Author Response

# More elaborations are preferred on the methodology of table 1, using the PRISMA method for example, when studies are limited to only 4 reviews or original studies, and this could be attached as supplementary materials.

Thank you very much for this comment. As we did not perform a systematic literature review, we did not use the PRISMA method. However, following part was added as supplementary material:

“-Search strategy on largest (n > 3000) meta-analyses or prospective studies reporting on bacterial co-/superinfection rates in COVID-19 patients:

Search strategies were applied in the PubMed database.

Search method: (("covid 19"[MeSH Terms]) OR "sars cov 2"[MeSH Terms]) AND ("co infection" OR "bacterial pneumonia" OR "superinfection")).

Following papers were excluded: papers including less than 3000 participants, letters and comments, case series, unrelated papers, interventional studies.

Searched for on October 10th 2022 (selection criteria: review, systematic review, meta-analysis or clinical trial): 140 results. Four meta-analyses and one prospective original paper were retained.”

# In table 1, another column deemed important to mention the type of study, i.e., either original or literature review.

The type of study is now mentioned in the first column.

# In table 1, in Russel et al study, it was not mentioned the rate of co-infection or superinfection. This cell should be unified with others. 

This is because Russell et al. doesn't provide exact rates of co- and superinfection. They state that of the 48.902 patients admitted to hospital with COVID-19, 1107 patients were recorded to have COVID-19 related respiratory or bloodstream infection (this is 2.2%). 70% of these infections were ‘secondary infections’. Based on those numbers, I determined the rates of co-infection and superinfection, which are respectively 0.7% and 1.5%. This was added to table 1.

# I would love to read what other nations in the east have done regarding this problem, especially in China, as the pandemic originated there.

Thank you very much for this great suggestion. Our perspective paper mainly focused on the European setting, as we believe that achieving uniform therapeutic COVID-19 guidelines is already a big challenge. We haven’t found any specific paper discussing antimicrobials stewardship or evolution of antibiotic use for bacterial co-/superinfections in China (we only know from patient series that almost 100% of hospitalized patients received antibiotics in China from the start of the pandemic. However, we did not found any comparative study to investigate the evolution of antibiotic use over time. However, we believe that comparing different regions (Europe, North America – China – Middle East – Southeast Asia) regarding antibiotic use at given time points in the COVID-19 setting would be of great value and we will consider this as potential future work.

# The part about variation in the definitions of bacterial coinfection and superinfection seems very interesting to read. It could be completed by adding a table of what international and trusted health officials (e.g., WHO. CDC. ECDC.. etc) have defined those terms. 

Thank you very much for this suggestion. When consulting those guidelines, we notice that the WHO and ECDC therapeutic guidelines don’t define bacterial co- and superinfection. The CDC did not publish any inpatient therapeutic guidance in the COVID-19 setting. However, this strengthens the urgent need for clear definitions of bacterial co-/superinfection in the scientific literature and guidelines. We added following sentence: “Moreover, international COVID-19 therapeutic guidelines such as the World Health Organization and the European Centre for Disease Control and Prevention guidelines, don’t define bacterial co-infection and superinfection [10]”.

# Figure 2: the title is a repetition of the footnotes and either one might be enough, and the figure could be moved to the supplementary materials as it is complementary to Table S1. 

We agree. The figure was moved to ‘Supplementary Materials’ and we deleted the repetitive part.

# The manuscript needs to be revised as it has some minor grammar and spelling mistakes, (for example: see line 143 "admisson >> admission??) 

Thank you very much. The manuscript has now been revised for minor grammar and spelling mistakes.

Reviewer 2 Report

The authors point out two points: the heterogeneity of the definitions used to identify co-infections and the variation in guidelines for the diagnosis/treatment of these co-infections. These are important and valid points.

However, the article would benefit from a clear structure. As a reader, it would be nice if the above points/purpose of the article were introduced early in the article (it now only comes at the bottom of page 2 and in somewhat vague terms). In addition, the introduction is now in parts too long.

The authors write in several places that PCT is useful in ruling out co-infections and refer to references 11 and 12, but these do not support this. Reference 11 says just the opposite and reference 12 does not address covid.

The conclusion also states that disease severity is a good predictor, but there is no substantiation there and probably this is just an association and does not distinguish well on its own.

Author Response

The authors point out two points: the heterogeneity of the definitions used to identify co-infections and the variation in guidelines for the diagnosis/treatment of these co-infections. These are important and valid points.

However, the article would benefit from a clear structure. As a reader, it would be nice if the above points/purpose of the article were introduced early in the article (it now only comes at the bottom of page 2 and in somewhat vague terms). In addition, the introduction is now in parts too long.

Thank you very much for this great remark. Although this paper was first designed as a ‘letter to the editor’, which is not a valid category in ‘Antibiotics’ and was thus changed to a ‘perspective article’, we agree that our paper would benefit of a clear structure. This is why we have now structured this paper in different parts. We also explain this in the introduction: ‘In this perspective article, we first provide a summary of the impact of the COVID-19 pandemic on antimicrobial stewardship efforts, AMR and the growing knowledge on bacterial co-/superinfection epidemiology and antibiotic (over)use in this context. Second, we offer a critical analysis of the major papers reporting on bacterial co-/superinfection (bCS) rates in COVID-19 patients. Last, we discuss the variation in European guidelines for the diagnosis/treatment of these bCS.’

The authors write in several places that PCT is useful in ruling out co-infections and refer to references 11 and 12, but these do not support this. Reference 11 says just the opposite and reference 12 does not address covid.

Thank you for this remark. We agree for reference 12. Reference 11 (May et al) indeed state that procalcitonin (cutoff, 0.25 or 0.50 ng/ml) do not reliably identify bacterial coinfections. However, they found high negative predictive values (more than 90%) for bacterial respiratory co-infection, making PCT a potential good excluder for bacterial co-infections. We replaced those two references by two papers of Dolci et al. and Vaughn et al. in which the authors underline the high negative predictive value of procalcitonin levels for bacterial co-infection in the COVID-19 setting.

The conclusion also states that disease severity is a good predictor, but there is no substantiation there and probably this is just an association and does not distinguish well on its own.

You are right. We did not imply any causal association with this statement. However, we do not want to confuse the reader and deleted ‘disease severity’.

Reviewer 3 Report

1.  Lines 41-43 require a reference.  authors need to review. 

2.  line 90 & 93 should have Mycoplasma pneumoniae italized.  Authors need to review. 

3.  unsure if the authors want more to be accomplished or want better guidelines.  Authors need to address. 

Author Response

1. Lines 41-43 require a reference.  authors need to review. 

Thank you for this comment. The reference of lines 41-43 is the same as the reference provided for line 44. This is now more clear.

2. line 90 & 93 should have Mycoplasma pneumoniae italized.  Authors need to review.

This was changed.

3. unsure if the authors want more to be accomplished or want better guidelines.  Authors need to address. 

In the conclusion, we plead for a uniform approach when reporting on bCS. Furthermore, we also plead for the implementation of clear international guidelines, which should be based on the latest evidence based medicine and knowledge, without being to ‘general’.

We changed following part:

“We therefore strongly advocate for the implementation of international diagnostic guidelines, using predictors and excluders of bCS and standardized definitions of bCS. These definitions and guidelines should be dynamic, could indicate the level of diagnostic certainty, and should evolve according to the best available evidence.”

To:

“We therefore strongly advocate for the implementation of international diagnostic guidelines, using predictors and excluders of bCS and standardized definitions of bCS. These definitions and guidelines should be dynamic and more detailed. Guidelines should be based on different clinical situations and could indicate the level of diagnostic certainty, and should evolve according to the best available evidence.”

Round 2

Reviewer 2 Report

The authors have addressed my suggestions

Reviewer 3 Report

I agree with the authors edits and believe the manuscript is appropriate for publication.